# Modified *Urtica dioica* Leaves as a Low-Cost and Effective Adsorbent for the Simultaneous Removal of Pb(II), Cu(II), Cd(II), and Zn(II) from Aqueous Solution

**DOI:** 10.3390/ijms26062639

**Published:** 2025-03-14

**Authors:** Enkhtuul Mendsaikhan, Munkhpurev Bat-Amgalan, Ganchimeg Yunden, Naoto Miyamoto, Naoki Kano, Hee Joon Kim

**Affiliations:** 1Graduate School of Science and Technology, Niigata University, 8050 Ikarashi 2-Nocho, Nishi-ku, Niigata 950-2181, Japan; f22k503k@mail.cc.niigata-u.ac.jp; 2Department of Chemical Engineering, School of Applied Sciences, Mongolian University of Science and Technology, 8th khoroo, Baga Toiruu 34, Sukhbaatar District, Ulaanbaatar 14191, Mongolia; munkhpurev@must.edu.mn (M.B.-A.); ganchimeg.yu@must.edu.mn (G.Y.); 3Department of Chemistry and Chemical Engineering, Faculty of Engineering, Niigata University, 8050 Ikarashi 2-Nocho, Nishi-ku, Niigata 950-2181, Japan; nmiyamoto@eng.niigata-u.ac.jp; 4Department of Environmental Chemistry and Chemical Engineering, School of Advanced Engineering, Kogakuin University, 2665-1 Nakano, Tokyo 192-0015, Japan; kim@cc.kogakuin.ac.jp

**Keywords:** *Urtica dioica* leaves (UDLs), multi-component heavy metals, removal efficiency, wastewater treatment, low-cost adsorbent

## Abstract

This study investigates the simultaneous adsorption of Pb(II), Cu(II), Cd(II), and Zn(II) ions from aqueous solutions using *Urtica dioica* leaves (UDLs) modified with sulfuric acid, followed by heat treatment to enhance adsorptive properties. The heat treatment significantly increased the adsorbent’s specific surface area to 451.93 m^2^·g^−1^. Batch adsorption experiments were performed to determine the influence of the contact time, pH of the aqueous solution, adsorbent dosage, temperature, and initial metal concentration on the adsorption efficiency. The material (modified UDLs) was characterized using X-ray diffraction (XRD), Fourier transform infrared spectroscopy (FT-IR), scanning electron microscope (SEM), and X-ray photoelectron spectroscopy (XPS). Maximum removal efficiencies were determined as 99.2%, 96.4%, 88.7%, and 83.6% for Pb(II), Cu(II), Cd(II), and Zn(II) ions, respectively. Adsorption isotherms and kinetics revealed that the process follows the Langmuir equation and pseudo-second-order models, indicating monolayer adsorption and chemisorption mechanisms. Furthermore, thermodynamic analysis indicated that the adsorption processes are spontaneous and endothermic in nature. The influence of competing ions on the adsorption of multiple heavy metals was also discussed. The results suggest that sulfuric acid and heat-treated *Urtica dioica* leaves can offer a promising, low-cost, and eco-friendly adsorbent for removing heavy metal ions from contaminated water.

## 1. Introduction

Water pollution caused by heavy metals has become a major environmental issue in recent years. Various industrial activities, including mining, electroplating, leather tanning, metal processing, steel metallurgy, pigment synthesis and dyeing, and battery production, release heavy metals into aquatic ecosystems [1,2,3]. Lead (Pb(II)), copper (Cu(II)), cadmium (Cd(II)), and zinc (Zn(II)) are some of the most common heavy metal pollutants found in industrial wastewater [4,5]. They are non-biodegradable and toxic pollutants that can accumulate in living organisms and the environment, leading to severe health problems like neurological, cardiovascular, carcinogenic, kidney, blood, nervous, and bone diseases [5,6,7]. Although numerous technologies, including chemical precipitation, ion exchange, ultrafiltration, reverse osmosis, membrane, and adsorption, have been employed to remove heavy metals from industrial wastewater, many developing countries find these modern solutions financially inaccessible [8,9,10,11,12]. Intending to combat these issues, research efforts have increased to develop an efficient, eco-friendly, and low-cost adsorbent to treat wastewater. Adsorbents can be used naturally or prepared in multiple ways [13]. Generally, many materials, such as activated carbon, different clays, polymeric materials, nanocomposites, and natural adsorbents, are used for adsorption [14,15]. Natural adsorbents such as various plant leaves [16,17,18], roots, seeds, tree barks, coconut shells, eggshells [19], tea, and agricultural wastes [13,20] have advantages such as very low costs compared to other adsorbents, are grown in many areas in nature, are easily supplied, are considered to produce zero waste, have a relatively high surface area, and have many functional groups [21]. Over the years, various methods including mechanical, thermal, and chemical treatments have been developed to enhance the adsorption capabilities of lignocellulosic biomass, broadening their applicability in heavy metal separation processes [22,23,24]. Sulfuric acid modification is well-suited for enhancing adsorbents because it effectively alters the chemical and structural properties of biomass, significantly boosting its adsorption capabilities. Additionally, sulfuric acid is an inexpensive reagent, making it a cost-effective choice for large-scale applications [25,26]. So far, we have studied some materials to enhance the adsorption capacity of heavy metals [27,28,29,30,31]. In this study, we chemically modified *Urtica dioica* leaves (UDLs) using sulfuric acid. Furthermore, these chemically modified UDLs underwent thermal treatment to produce the final adsorbent material with enhanced adsorption properties.

*Urtica dioica*, usually called common nettle, belongs to the Urticaceae family and grows worldwide, with the ability to grow 1 to 2 m. The main components of *Urtica dioica* are cellulose (80.1–90.4%), hemicellulose (5.6–9.5%), and lignin (1.4–4.4%) [32]. These constituents have potential functional groups, such as hydroxyl, carbonyl, amino, carboxylic, and alkoxyl, with good affinities for metal ions. Likewise, *Urtica dioica* has a significant adsorption potential, as it is soft, resistant, and has a low specific weight. In the literature, it is also known that this plant is used for phytoremediation [33,34]. In several studies, the authors used *Urtica dioica* as a low-cost adsorbent for removing pollutants from an aqueous solution [35,36,37,38]. Moreover, it was found that chemically treated biomass adsorbents exhibit significantly higher adsorption capacities compared to untreated ones [39]. Abundant natural materials have the potential to be used as inexpensive sorbents. Furthermore, when these materials reach the end of their lifetime, they can be disposed of without needing costly regeneration.

Polluted effluent water, with concentrations above the regulated limit, contains a myriad of metal ions and, as such, mono- or bi-adsorption systems are not realistic; hence, simultaneous multi-metal adsorption systems [40,41] are needed. The concentration range of 50–500 mg/L of for Pb(II), Cu(II), Cd(II), and Zn(II) chosen in this work can be found in untreated industrial wastewater, such as from mining, battery manufacturing, electronics recycling, and metal finishing [42,43]. Altering the ratios of metal ions in solution can significantly impact adsorption behavior due to competition for active sites on the adsorbent. For instance, in a study examining the competitive adsorption of Cd^2+^, Pb^2+^, and Cu^2+^ ions using a multigroup-functionalized cellulose adsorbent, it was observed that Cu^2+^ exhibited the strongest inhibitory effect on the adsorption of the other metals. This indicates that higher concentrations of Cu^2+^ can suppress the adsorption of Cd^2+^ and Pb^2+^ [44]. In our study, the chosen metal ion ratios reflect typical mixtures in industrial wastewater, allowing the observation of competitive adsorption between metals. The present research is focused on the behavior of modified *Urtica dioica* leaves regarding multi-component heavy metals Pb(II), Cu(II), Cd(II), and Zn(II), evaluating different parameters in the system, such as pH, particle size, biosorbent dosage, time, and metal ion concentration. The removal efficiency of the multi-metal ion mixtures was discussed. The characterization of the adsorbents was performed using Fourier transform infrared spectroscopy (FTIR), X-ray photoelectron spectroscopy (XPS), scanning electron microscopy (SEM-EDS), and X-ray diffraction (XRD) analyses. The equilibrium data were fitted to Langmuir and Freundlich isotherm models, while the kinetics data were correlated according to typical kinetics models (pseudo-first-order and pseudo-second-order model).

The main original points in this paper are (1) effectively utilizing *Urtica dioica* leaves as a low-cost and eco-friendly adsorbent for the simultaneous removal of multi-component heavy metals, and (2) determining the activation energy (not only thermodynamic parameters such as $\Delta G^{0},\;\Delta H^{0},\;\mathrm{and}\;\Delta S^{0}$) to quantify the energy barrier of the adsorption process.

## 2. Results and Discussion

### 2.1. Characterizations of Materials

#### 2.1.1. SEM-EDS Analysis

SEM images for unmodified UDLs (a), H_2_SO_4_-modified UDLs (b), and H_2_SO_4_-modified UDLs after adsorption (c) along with TUDL500 (d,e) are presented in Figure 1. To enhance imaging, the samples were coated with a thin layer of gold. As shown in Figure 1a, unmodified UDLs had a rough and irregular surface structure. Figure 1b shows the increased surface roughness and the appearance of micropores, which improve adsorption properties. After the adsorption of heavy metals, the interaction of H_2_SO_4_-modified UDLs with heavy metals resulted in the formation of white, shiny deposits on its surface, as illustrated in Figure 1c. The SEM image after heat treatment (Figure 1d) reveals a significant increase in porosity, with interconnected pores forming a network. It is suggested that this structure enhances adsorption capacity by increasing the surface area. Figure 1e reveals a highly porous structure, a characteristic feature retained from the pyrolysis process. However, many of the visible pores in the image are partially filled, providing direct evidence of heavy metal ion adsorption. The EDS analysis is presented in the Appendix A.

#### 2.1.2. N_2_-BET Analysis

The N_2_-BET analysis (Table 1) shows that chemical modification and pyrolysis significantly improve the adsorption properties of UDLs. The unmodified UDLs exhibit a low specific surface area (0.4659 m^2^·g^−1^), which increases slightly after H_2_SO_4_ modification (1.0157 m^2^·g^−1^). Pyrolysis at 500 °C (TUDL500) dramatically enhances the specific surface area (451.9304 m^2^·g^−1^) and creates a microporous structure, making it effective for heavy metal adsorption.

#### 2.1.3. XPS Spectra

Unmodified UDLs (a), H_2_SO_4_-modified UDLs (b), and those after the adsorption of heavy metals (c) were analyzed using XPS, as shown in Figure 2 and Table 2. In the unmodified UDLs, the primary elements detected were carbon (81.04%), nitrogen (2.79%), oxygen (13.74%), and a small presence of calcium (1.59%). Upon modification with sulfuric acid, significant changes in surface chemistry were observed: sulfur appeared at 1.4%, and calcium increased to 3.38%. This indicates the successful incorporation of sulfonate and calcium groups, which is consistent with the sulfonation reaction expected from H₂SO₄ treatment.

After adsorption, XPS spectra showed peaks for Pb4f, Cu2p, Cd3d, and Zn2p, confirming heavy metal binding to the modified UDL surface (Figure 3). The reduction in sulfur (from 1.4% to 0.35%) and calcium (from 3.38% to 0.13%) suggests their involvement in metal binding, likely via ion exchange. The data indicate that adsorption occurs through multiple mechanisms: chemisorption, evidenced by binding energy shifts and strong interactions with sulfonate and hydroxyl groups; ion exchange, as shown by the displacement of sulfur and calcium; and complexation, supported by an increased oxygen content facilitating metal–ligand formation.

In addition, XRD patterns are presented in the Appendix A.

#### 2.1.4. FT-IR Spectra

The FT-IR spectrum of unmodified UDLs, H_2_SO_4_-modified UDLs, and after the adsorptions of heavy metals is shown in Figure 4. In the spectrum, the broad and strong bands at 3200–3500 cm^−1^ are attributed to hydroxyl (O–H) groups of the biomass. The 2927 and 2853 cm^−1^ peaks correspond to the aliphatic group (C–H, CH_2_) stretching vibrations [45]. The peak at 1646 cm⁻^1^ indicates the C=O stretching vibration of carboxylic acid or other carbonyl-containing groups [46]. The observed peak at 1422 cm^−1^ shows the presence of calcite (CaCO_3_) [47]. The broad peak at 1059 cm^−1^ is assigned to (C–O, CC, CH_2_) from lignin or hemicellulose [48]. After modification, peak intensities at 3200–3500 cm^−1^ are decreased due to degradation of the hydrogen bond between the cellulose chain during the hydrolysis process. Moreover, new peaks at 1162 cm^−1^ (the glucopyranoside ring stretching), 675 cm^−1^, and 615 cm^−1^ (S=O) functional groups were formed [49]. The peak at 1380 cm⁻¹ corresponds to the symmetric bending vibrations of methyl groups (CH₃), characteristic of aliphatic hydrocarbons. This sharp peak suggests the presence of organic components in *Urtica dioica*, which may be modified during treatment. While CH₃ groups do not directly bind heavy metals, changes in intensity or shape after adsorption may indicate structural alterations in nearby functional groups [50]. After the adsorption of Pb(II), Cu(II), Cd(II), and Zn(II), the peaks at 3362 cm^−1^ (−OH and −COOH groups), 2927 cm^−1^ (−CH_3_), and 1646 cm^−1^ (C=O or −COOH) decreased due to the interaction between the heavy metal molecules and functional groups on the surface of the modified adsorbent [51,52]. Meanwhile, the area of these peaks at 1422 cm^−1^ (CaCO_3_), 1162 cm^−1^ (the glucopyranoside ring stretching), 1059 cm^−1^ (C–O, CC, CH_2_), 675 cm^−1^, and 615 cm^−1^ (S=O) disappeared completely, demonstrating that these groups also participated in metal adsorption [45,51,53]. The 400–540 cm⁻¹ peaks indicate the formation of M-O bonds resulting from interactions between metal ions (Pb(II), Cu(II), Cd(II), Zn(II)) and oxygen-containing functional groups (hydroxyl, carboxyl, sulfate) on the adsorbent surface. Moreover, the FTIR spectra of H_2_SO_4_-modified UDLs (with various concentrations of H_2_SO_4_) are presented in the Appendix A in more detail.

### 2.2. Evaluation of Multi-Heavy-Metal Adsorption Performance

#### 2.2.1. Effect of Sulfuric Acid Concentration

To determine the modification effect of the sulfuric acid concentration on the adsorption of multiple heavy metals, the sulfuric acid concentration was changed (0–98%), and other conditions remained unchanged (temperature: 298 K, pH: 6, contact time: 2 h, adsorbent dosage: 2 g/L, initial concentration: 50 mg/L). The results are shown in Figure 5. The figure shows that the chemically modified UDLs were able to absorb higher amounts of Pb(II), Cu(II), Cd(II), and Zn(II) than the unmodified UDLs. For the acid concentration of 40 -H_2_SO_4_, more than 90% of all four heavy metals was adsorbed simultaneously, and when the acid concentration was greater than 40% H_2_SO_4_, the adsorption amount decreased slowly. This phenomenon may occur because, when exceeding concentrations of 40% H_2_SO_4_, the adsorbent can become over-processed, leading to the deterioration or loss of functional groups, as depicted in Appendix A. Additionally, excessive acid might damage the surface structure or reduce the availability of active sites for metal binding, ultimately decreasing adsorption efficiency [54]. Therefore, in this study, a 30% H_2_SO_4_ concentration was chosen for the following experiments.

#### 2.2.2. Effect of Initial pH

The pH value of the aqueous solution can influence the surface charge and metal-binding sites of the adsorbent, as well as the ionization state and form of metal ions [55]. Generally, under high-pH conditions, multiple heavy metal ions form hydroxide precipitates. Therefore, in this experiment, pH tests were performed in the pH range of 1–6. The other adsorption conditions were the same as those in the former (2.2.1) experiment. As shown in Figure 6, the removal efficiency (%) increases with the pH value and reaches a maximum around pH 6. The removal efficiencies at pH 6 were determined as 99.2%, 95.3%, 83%, and 71.6% for Pb(II), Cu(II), Cd(II), and Zn(II) ions, respectively. Therefore, further adsorption experiments were conducted at pH 6. In this study, the zeta potential of H₂SO₄-modified UDLs was measured to investigate the surface charge of the material, and the results are shown in the Appendix A.

Effects of the adsorbent time and dosage on multi-heavy-metal adsorption performance are also presented in the Appendix A.

#### 2.2.3. Effect of Heat Treatment

Figure 7 shows the effect of calcination temperatures (300 °C, 400 °C, and 500 °C) on the adsorption capacities of Pb(II), Cu(II), Cd(II), and Zn(II) using H_2_SO_4_-modified UDLs. The results indicate that increasing the pyrolysis temperature enhances the adsorption capacities for all metals. The adsorption capacities were 46.6 mg/g, 37.1 mg/g, 11.9 mg/g, and 10.6 mg/g for Pb(II), Cu(II), Cd(II), and Zn(II), respectively, at 500 °C. 

#### 2.2.4. Effect of Competitive Ions

The effect of competitive ions on multi-heavy-metal adsorption is shown in Figure 8. In this experiment, the adsorption of multiple heavy metals was investigated under the presence of solutions with different concentrations (i.e., 0, 10, 25, 50, and 100 ppm) of each individual (Na^+^, K^+^, Ca^2+^, Mg^2+^) and their complex (Na^+^ + K^+^ + Ca^2+^ + Mg^2+^). The diagram shows that the adsorption capacities of modified UDLs for multiple heavy metals decrease slightly as the concentration of competing ions increases. However, no notable decrease in the adsorption capacity was observed. This shows that even with a high concentration of competing ions, modified UDLs also show a good adsorption capacity for multiple heavy metals and can be used as an effective adsorbent [55]. Lead (Pb^2+^) ions, with a larger ionic radius and lower hydration energy than zinc (Zn^2+^) ions, are more easily attracted to the adsorbent’s surface, readily occupying adsorption sites. This is due to their easier dehydration and higher affinity for functional groups (e.g., carboxyl, hydroxyl) on the modified adsorbent compared to Cu^2+^, Cd^2+^, and Zn^2+^ ions [56,57].

### 2.3. Adsorption Kinetics Study

Kinetic investigation of the adsorption process reveals the rate and mechanism of the reaction. These were studied by fitting the experimental data to the pseudo-first-order (PFO) reaction equation and pseudo-second-order (PSO) reaction equation in their linear and non-linear forms. The kinetic equations are described in more detail in adsorption kinetic and isotherm models in Appendix A.

The influence of the contact time on heavy metal adsorption onto modified UDLs was assessed because the adsorption rate and kinetics are crucial for batch-biosorption experiments. The effect of the contact time in the range of 0–15 h for the adsorption of Pb(II), Cu(II), Cd(II), and Zn(II) on the surface of modified UDLs was investigated at a constant pH (=6), constant dosage (2 g/L), constant initial concentration (50 mg/L), and in the range of 298 K. The results are shown in Figure 9 and Table 3, as well as Appendix A. The uptake was quick at the beginning because there are many accessible binding sites on the surface of modified UDLs. The equilibrium state was reached at 3 h for heavy metal adsorption. Since the removal efficiencies did not change much after the equilibrium state, further adsorption experiments were conducted for heavy metals at this contact time (3 h).

In Table 3, it is demonstrated that the PSO model generally exhibited superior correlation coefficients compared to the PFO model, although the PFO model is also fit for Pb(II). Additionally, the equilibrium adsorption capacities (*q_e_*) predicted by the PSO model were in closer agreement with the experimental values for all metal ions, indicating that chemisorption is the predominant rate-controlling mechanism.

### 2.4. Adsorption Isotherm Study

Adsorption isotherms are necessary to understand the adsorption process and predict the adsorption behavior of pollutants onto the adsorbent surface. To evaluate the adsorption data, Langmuir and Freundlich isotherm models were used.

According to the Langmuir isotherm model, the surface of the adsorbent is uniform. During adsorption, each surface molecule or atom of the adsorbent adsorbs a gas molecule, and the gas molecules adsorb on the solid surface as a single layer. There is no force between the gas molecules. This is provided by the following equation:(1)$$\frac{C_{e}}{q_{e}}=\frac{C_{e}}{q_{max}}+\frac{1}{K_{L}q_{max}}$$
where *C_e_* and *q_e_* are the heavy metal concentration (mg·L^−1^) and adsorption capacity (mg·g^−1^) when the adsorption reaches equilibrium, *q_max_* is the maximum adsorption capacity of the adsorbent (mg·g^−1^), and *K_L_* is the adsorption constant of the Langmuir isotherm (L·mg^−1^). The relationship between *C_e_*/*q_e_* and *C_e_* gives a straight line with a slope of 1/*q_max_* and intercept of 1/(*K_L_q_max_*).

Freundlich’s isotherm model considers multi-layer adsorption without considering the adsorption saturation on multi-layer heterogeneous surfaces. The isotherm of the linear Freundlich model is represented by Equation (4):(2)$$lnq_{e}=lnK_{F}+\frac{1}{n}lnC_{e}$$

Among them, *K_F_* is the adsorption capacity ((mg·g^−1^)·(dm^−3^·mg^−1^)^1/*n*^), and 1/*n* is the adsorption strength. The relationship between *lnq_e_* and *lnC_e_* can be used to determine the 1/*n* and *K_F_* values. The value of 1/*n* can be used to judge the difficulty of the adsorption process: irreversible adsorption (1/n=0), favorable adsorption (0<1/n<1), and unfavorable adsorption (1/n>1).

Under the optimal adsorption conditions (pH 6, temperature of 298 K, contact time of 3 h, and adsorbent dosage of 2 mg·L^−1^), the adsorption isotherm model of modified UDLs for multiple heavy metals was established at the initial concentration of 10–500 mg·L^−1^. In case of the Langmuir model and Freundlich model (Figure 10), linear correlation coefficients (*R*^2^) were determined (Table 4). In this study, we performed error analyses [58], and the results are presented in Table 4. Specifically, χ^2^, MPSD, and RMSE values were calculated. The correlation coefficients (*R*^2^) showed that the Langmuir isotherms and Freundlich isotherms could describe the adsorption of multiple heavy metals well. Although both Langmuir isotherm models and Freundlich isotherm models could express the adsorption of multiple heavy metals on modified UDLs well, the correlation coefficient of the Langmuir isotherm models was higher, indicating that the adsorption of multiple heavy metals by modified UDLs was more in line with the Langmuir isotherm models. The results showed that the adsorption of multiple heavy metals by modified UDLs mainly occurred through a single-layer reaction. Based on the model, the maximum adsorption capacities were estimated to be 98.3 mg·g^−1^ (Pb(II)), 73.4 mg·g^−1^ (Cu(II)), 12.9 mg·g^−1^ (Cd(II)), and 13.9 mg·g^−1^ (Zn(II))).

We also determined non-linear Langmuir and Freundlich isotherms of multi-heavy-metal adsorption, as detailed in the Appendix A.

### 2.5. Adsorption Thermodynamics

Thermodynamic tests were performed with the temperature changing between 298 K and 328 K. This study was conducted by changing the temperature (298, 308, 318, 328 K) and keeping all other parameters unchanged. The results are shown in Figure 11. Maximum removal efficiencies at 318 K were determined as 99.2%, 96.4%, 88.7%, and 83.6% for Pb(II), Cu(II), Cd(II), and Zn(II) ions, respectively. Therefore, 318 K was chosen as the most suitable temperature for this study.

The thermodynamic parameters for adsorption, including the changes in Gibb’s standard free energy $\Delta G^{0}$ (kJ·mol^−1^), standard enthalpy $\Delta H^{0}$ (kJ·mol^−1^), and entropy $\Delta S^{0}$ (J·(mol·K)^−1^), were determined (Table 5) using the van Hoff equation [59,60].(3)$$\Delta G^{0}=-RTlnK_{c}$$(4)$$lnK_{c}=-\frac{\Delta H^{0}}{RT}+\frac{\Delta S^{0}}{R}$$
where *R* is the universal gas constant (8.314 J·(mol·K)^−1^) and *T* is the temperature (K). The value of $K_{c}$ can be obtained by $K_{c}=q_{e}/C_{e}$; $q_{e}$ is the equilibrium adsorption capacity, mg·g^−1^; and $C_{e}$ is the equilibrium concentration of Pb(II), Cu(II), Cd(II), and Zn(II), mg·L^−1^. The slope and intercept of the linear relationship between $lnK_{c}$ and 1/*T* in Equation (4) were used to calculate $\Delta H^{0}$ and $\Delta S^{0}.$

Gibbs free energy (∆*G*^0^) has a negative value, and it tends to be a large negative value with an increasing temperature for all heavy metals, indicating that the adsorption reaction of heavy metals is spontaneous. Given the positive values of the change in enthalpy (∆*H*^0^), the adsorption process was endothermic for multi-component heavy metals. The positive values of the entropy (∆*S*^0^) show that irregularity was dominant on the adsorbent surface. From this table, it is suggested that an increased temperature provides additional energy to overcome the activation barrier for adsorption. This indicates that the adsorbent’s interaction with heavy metal ions involves energy-intensive processes, likely due to chemisorption, which requires bond formation between the adsorbate and the adsorbent surface. The Langmuir isotherm fit, which indicates monolayer adsorption, was elaborated on by discussing the specific properties of the adsorbent, such as uniform active sites on the UDL surface resulting from sulfuric acid modification [60]. We justified the pseudo-second-order kinetics by linking them to chemisorption mechanisms. This is consistent with the observed adsorption process involving electron sharing or exchange between functional groups on the adsorbent (e.g., sulfonic, calcium, or hydroxyl groups) and metal ions. The discussion also addresses the role of the adsorbent’s physical and chemical properties in driving this kinetic behavior.

A comparison of the adsorption capacities of the H_2_SO_4_-modified UDLs and TUDLs adsorbents with other previously reported adsorbents is presented in Table 6. As indicated in Table 6, the modified UDLs demonstrated an effective adsorption capacity for multi-component heavy metals compared to other adsorbents from earlier studies. A feature of this research is the identification of a simple and cost-effective method for producing adsorbents.

After calcination, the adsorption capacities did not increase in this study (Appendix A), although the specific surface area of the adsorbent significantly increased, suggesting that other factors might be influencing this outcome. One possible explanation is that heat treatment can damage or reduce the effectiveness of functional groups of H_2_SO_4_-modified UDLs. Therefore, further investigations will be conducted to evaluate the effects of other heat treatment conditions in more detail in future studies.

### 2.6. Chemical Mechanism

The FTIR analysis results confirmed that there were hydroxyl, carboxyl, sulfonate, and Ca^2+^ groups on the adsorbent surface, and the adsorption process was realized through weak interactions, such as electrostatic interactions and hydrogen bonds, through these functional groups. After the adsorption of Pb(II), Cu(II), Cd(II), and Zn(II), the peaks at 3362 cm^−1^ (−OH and −COOH groups), 2927 cm^−1^ (−CH_3_), and 1646 cm^−1^ (C=O) decreased due to the interaction between the heavy metal molecules and functional groups on the surface of the modified adsorbent. Meanwhile, the area of these peaks at 1422 cm^−1^ (CaCO_3_), 1162 cm^−1^ (the glucopyranoside ring stretching), 1059 cm^−1^ (C–O, C-C, CH_2_), 675 cm^−1^, and 615 cm^−1^ (S=O) disappeared completely, demonstrating that these groups also participated in metal adsorption. From the above results, we surmised that the functional groups (−OH, C−O, the glucopyranoside ring stretching, C=O, and S=O) played an essential role in the adsorption of metals through electrostatic interaction and surface complexation. The XRD analysis showed that calcium crystals were increased after modification, suggesting that calcium ions play a significant role in the adsorption process by participating in ion-exchange reactions during heavy metal adsorption. XPS analysis results indicated the successful adsorption of Pb(II), Cu(II), Cd(II), and Zn(II) by showing the appearance of corresponding peaks for these metals (e.g., Pb 4f, Cu 2p, Zn 2p, and Cd 3d) after adsorption. Furthermore, the analysis indicated that the adsorption mechanism involves ion exchange, as demonstrated by the reduction in calcium content on the adsorbent surface. This ion-exchange mechanism contributes to the efficient removal of heavy metals from aqueous solutions. According to the thermodynamic analysis results, the adsorption processes are spontaneous and endothermic. Overall, the kinetic studies show that the adsorption process included monolayer adsorption and chemisorption mechanisms.

## 3. Materials and Methods

All chemical reagents, including sulfuric acid, sodium hydroxide, and hydroxyl acid, used in this study were analytical grade and sourced from Kanto Chemical Co., Inc. (Tokyo, Japan). Stock solutions (1000 ppm) of Pb(II), Cu(II), Cd(II), and Zn(II) ions were prepared using Pb(NO_3_)_2_, Cu(NO_3_)_2_·3H_2_O, Cd(NO_3_)·4H_2_O, and Zn(NO_3_)_2_·6H_2_O in deionized water, respectively. These metal solutions with different concentrations (as required for our experiments) were made by diluting them conveniently. Ultrapure water with a resistivity of 18.2 MΩ cm (RFU 424TA, Advantech Aquarius, Suite A Dublin, CA, USA) was used in the experiments. In addition, a water bath incubator (BT100, Yamato Kagaku Co., Ltd., Tokyo, Japan), a vacuum drying oven (DP33, Yamato Kagaku Co., Ltd., Tokyo, Japan), and a pH meter (HORIBA F-72, Tokyo, Japan) were used in this work.

Selection of the plant, synthesis of the adsorbent, and thermal treatment of sulfuric acid-modified *Urtica dioica* are outlined in the Appendix A. The sample after heat treatment is abbreviated as “TUDL” in the following.

### 3.1. Characterization of the Adsorbent

The surface functional groups of the adsorbents were identified using Fourier Transform Infrared Spectroscopy (FTIR; FT-IR4200, Jasco, Hachioji, Tokyo, Japan). The FTIR spectra of chemically modified *Urtica dioica* leaves before and after adsorption were obtained in the frequency range of 400–4000 cm^−1^ using the KBr pellet method. X-ray diffraction (XRD) analysis was performed on an X-ray diffractometer (XRD; D2 Phaser, Bruker, Billerica, MA, USA) with Cu-Kα radiation, and the scanning test range was set to 10–70° (diffusion angle). The surface morphology of the materials was observed using scanning electron microscopy (SEM-EDS; JCM-6000 with JED-2300, JEOL, Akishima, Tokyo, Japan) and ion sputtering (JFC1100E, JEOL, Akishima, Tokyo, Japan). The morphologies of unmodified UDLs, modified UDLs, and those after the adsorption of heavy metals were studied using X-ray photoelectron spectroscopy (XPS, K-Alpha, Thermo Scientific Center, Waltham, MA, USA).

### 3.2. Adsorption Experiments

The adsorption efficiency of the Pb^2+^, Cu^2+^, Cd^2+^, and Zn^2+^ complex solutions with modified UDLs was determined using the following experiments. Initially, adsorption was performed by adding 0.1 g of adsorbent to 50 mL of complex heavy metal solution in a water bath equipped with a heater and shaker. The optimum conditions were determined by conducting adsorption experiments at a 1–6 initial pH, 0–24 h contact time, 0.05–0.20 g/50 mL adsorbent dosage, and 50–500 mg/L heavy metal concentration ranges. The pH of heavy metal solutions was adjusted with 0.1 N of HCl and 0.1 N of NaOH. In our experiments, we used a mixed cellulose ester membrane filter. The filtration time was approximately 3–7 s. The concentrations of heavy metals before and after adsorption in the heavy metal solution were analyzed using inductively coupled plasma atomic emission spectrometry (ICP-AES; SPS1500, SEIKO, Chiba, Japan).

The adsorption capacity, *q* (mg/g), was calculated using the following equation [27]:(5)$$q=\left(C_{o}-C_{e}\right)\cdot \frac{V}{m}$$
where *C*_0_ and *C_e_* are the initial and equilibrium heavy metal concentrations (mg·L^−1^), consecutively; *V* is the volume of solution (L); and *m* is the mass of the adsorbent used (g).

The removal efficiency (*R*) was calculated based on the following equation:(6)$$R\%=\frac{\left(C_{0}-C_{e}\right)}{C_{0}}\cdot 100$$

## 4. Conclusions

Modified *Urtica dioica* leaves (UDLs) were successfully developed as a low-cost and eco-friendly adsorbent for the simultaneous removal of Pb(II), Cu(II), Cd(II), and Zn(II) from aqueous solutions. The physicochemical characterization, including XRD, FTIR, SEM, and XPS, confirmed the structural modifications of UDLs, leading to an enhanced adsorption capacity. Sulfuric acid and heat treatment introduced sulfonate groups, increased calcium content, and improved the porosity and surface area of the adsorbent, which facilitated metal ion binding through electrostatic interactions and ion exchange. The higher correlation with Langmuir isotherms and pseudo-second-order kinetics indicates monolayer adsorption and predominant chemisorption mechanisms. Maximum removal efficiencies were determined as 99.2%, 96.4%, 88.7%, and 83.6% for Pb(II), Cu(II), Cd(II), and Zn(II) ions, respectively, with optimum conditions at pH 6, 318K, and a 2 g/L adsorbent dosage. Given the positive values of the change in enthalpy (∆*H*^0^), the adsorption process was endothermic for multi-component heavy metals. Overall, this study demonstrates that sulfuric acid and heat-treated UDLs are an efficient, cost-effective, and sustainable material for removing heavy metals from contaminated water. The findings suggest that they have potential in wastewater treatment, particularly in regions where affordable and accessible solutions are required.

## Figures and Tables

**Figure 1 ijms-26-02639-f001:**
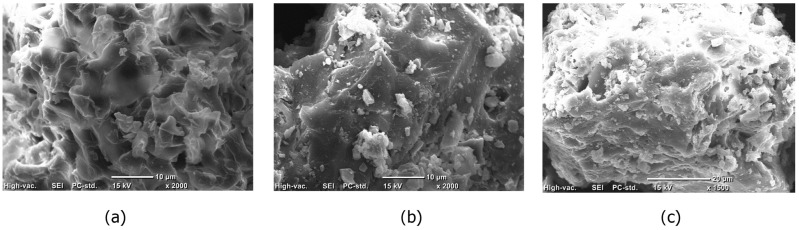
SEM images of unmodified UDLs (**a**), H_2_SO_4_-modified UDLs (**b**); H_2_SO_4_-modified UDLs after adsorption (**c**), and TUDL500 (**d**,**e**).

**Figure 2 ijms-26-02639-f002:**
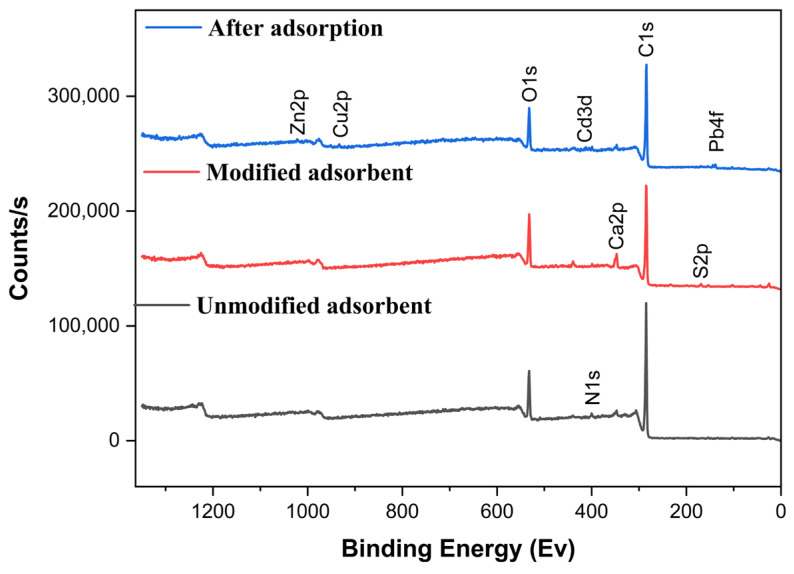
The XPS images of unmodified UDLs, H_2_SO_4_-modified UDLs, and after the adsorption of heavy metals.

**Figure 3 ijms-26-02639-f003:**
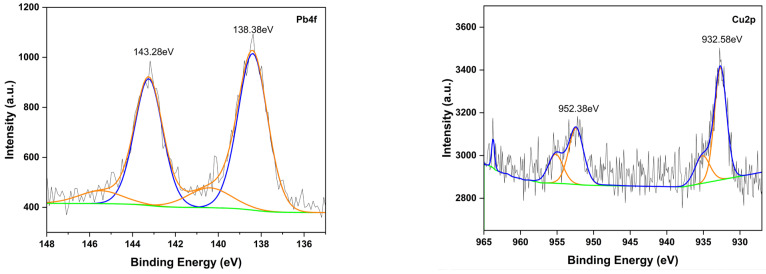
The XPS spectra after the adsorption of heavy metals. The colored lines represent the peak components obtained from peak fitting. In each spectrum, the blue and orange lines correspond to the deconvoluted peaks: Pb 4f₇/₂ and Pb 4f₅/₂ for Pb 4f, Cu 2p₃/₂ and Cu 2p₁/₂ for Cu 2p, Cd 3d₅/₂ and Cd 3d₃/₂ for Cd 3d, and Zn 2p₃/₂ and Zn 2p₁/₂ for Zn 2p.

**Figure 4 ijms-26-02639-f004:**
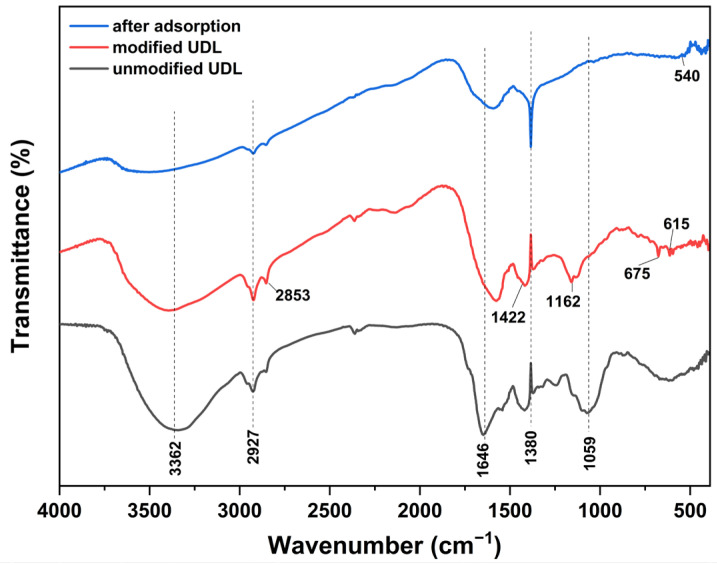
FT−IR of UDLs, H_2_SO_4_-modified UDLs, and those after the adsorption of heavy metals.

**Figure 5 ijms-26-02639-f005:**
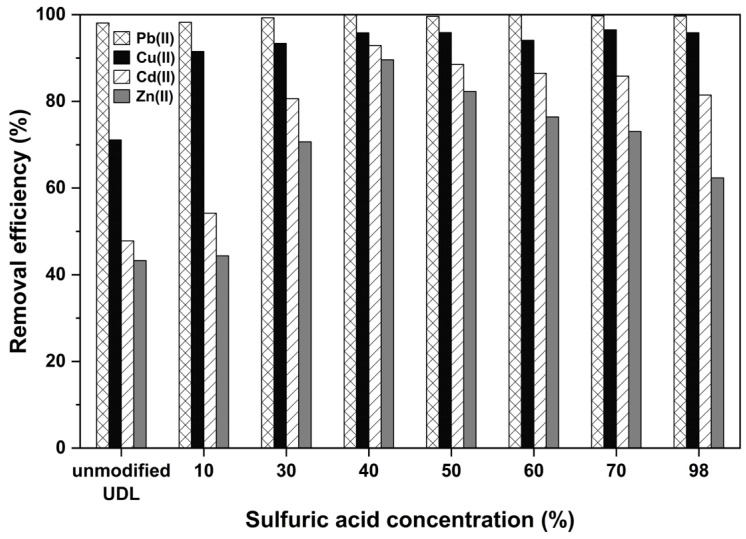
The effect of sulfuric acid concentration on the adsorption of multiple heavy metals.

**Figure 6 ijms-26-02639-f006:**
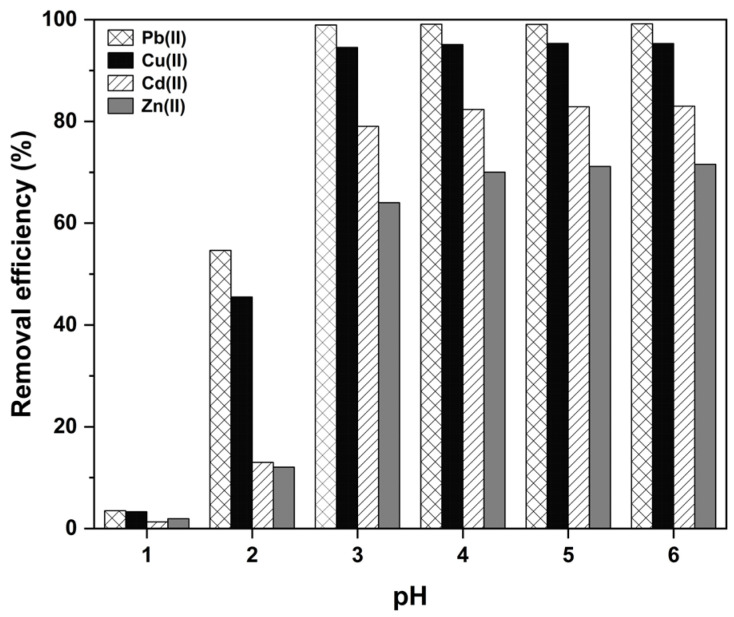
The effect of pH on the adsorption of multiple heavy metals by modified UDLs.

**Figure 7 ijms-26-02639-f007:**
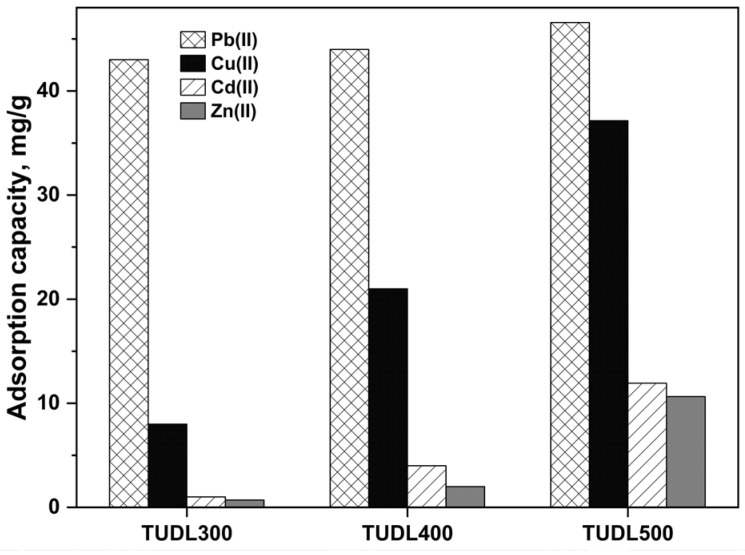
The effect of roasting temperatures on the adsorption of multiple heavy metals.

**Figure 8 ijms-26-02639-f008:**
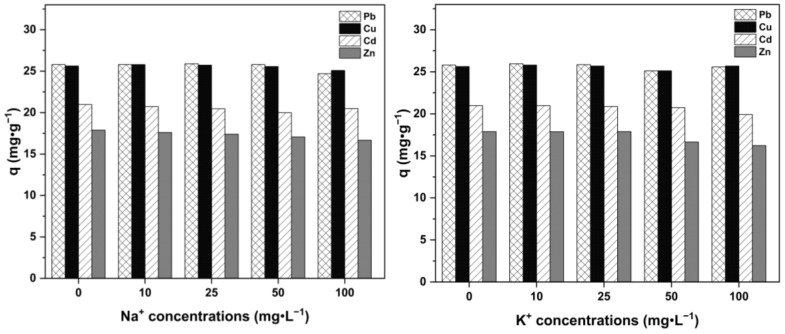
The effect of competitive ions on the adsorption of multiple heavy metals by modified UDLs.

**Figure 9 ijms-26-02639-f009:**
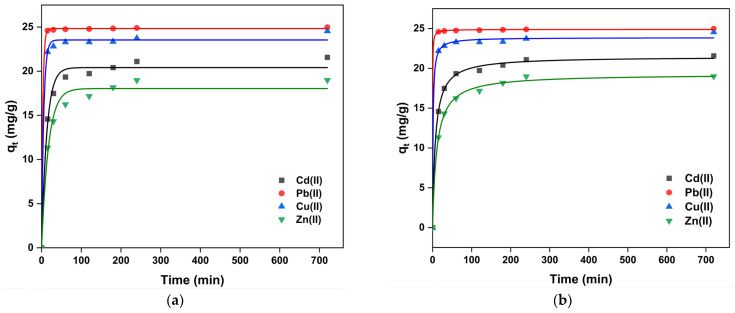
Pseudo-first-order non-linear kinetic models (**a**) and pseudo-second-order non-linear kinetic models (**b**) of multi-heavy-metal adsorption on the modified UDLs.

**Figure 10 ijms-26-02639-f010:**
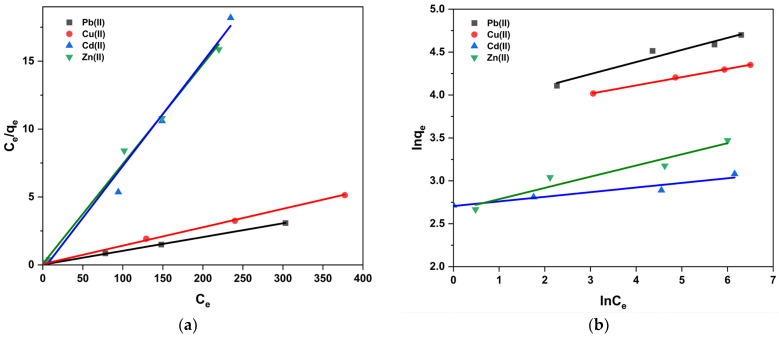
Linear Langmuir (**a**) and Freundlich (**b**) isotherms of multi-heavy-metal adsorption on modified UDLs.

**Figure 11 ijms-26-02639-f011:**
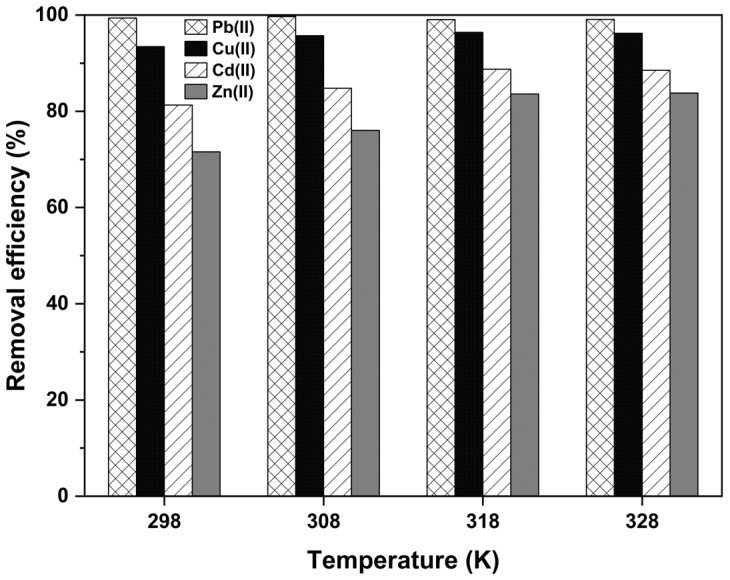
The effect of temperature on the adsorption of multiple heavy metals by modified UDLs.

**Table 1 ijms-26-02639-t001:** BET specific surface area, pore size, and pore volume of unmodified and H_2_SO_4_-modified UDLs.

UDLs	BET Specific Surface Area (m^2^·g^−1^)	Pore Volume (cm^3^·g^−1^)	Pore Size (nm)
Unmodified	0.4659	0.00092	7.89508
H_2_SO_4_-modified	1.0157	0.00397	15.63720
TUDL500	451.9304	0.252627	2.23598

**Table 2 ijms-26-02639-t002:** The chemical composition of UDLs determined using XPS.

UDLs	C	N	O	S	Ca
Unmodified (%)	81.04	2.79	13.74	-	1.59
Modified (%)	74.97	2.02	16.47	1.4	3.38
After adsorption (%)	80.46	2.88	14.19	0.35	0.13

**Table 3 ijms-26-02639-t003:** The parameters for the two kinetic non-linear models of adsorption of multiple heavy metals by modified UDLs.

	Pb(II)	Cu(II)	Cd(II)	Zn(II)
Pseudo-First-Order Model				
*q_e_* (mg·g^−1^)	24.83	23.54	20.41	18.04
*Κ*_1_ (min^−1^)	0.31	0.18	0.07	0.06
*R* ^2^	0.9999	0.9966	0.9882	0.9820
Pseudo-Second-Order Model				
*q_e_* (mg·g^−1^)	24.89	23.87	21.47	19.27
*Κ*_2_ (g mg^−1^ min^−1^)	0.20	0.04	0.007	0.005
*R* ^2^	0.9999	0.9983	0.9985	0.9979
*q_e exp_* (mg·g^−1^)	25.52	25.64	22.52	20.90

**Table 4 ijms-26-02639-t004:** The coefficient of linear Langmuir and Freundlich isotherms for multiple heavy metals.

	Pb(II)	Cu(II)	Cd(II)	Zn(II)
Langmuir Equation				
*q_max_* (mg·g^−1^)	98.72	73.85	13.06	13.61
*Κ_L_* (L·mg^−1^)	0.0002	0.001	0.028	0.006
*R* ^2^	0.99	0.99	0.98	0.99
χ^2^	0.03	0.26	0.01	0.02
MPSD	0.007	0.05	0.12	0.01
RMSE	1.15	3.22	0.04	0.37
Freundlich Equation				
*R* ^2^	0.93	0.99	0.63	0.63
1/*n*	0.14	0.10	0.05	0.13
*Κ_F_* (mg·g^−1^)	45.69	41.54	14.97	14.26
χ^2^	60.55	24.51	0.28	0.01
MPSD	28.47	19.43	2.68	0.07
RMSE	1500.21	986.15	4.01	0.03
*q_max,exp_* (mg·g^−1^)	98.29	73.45	12.90	13.87

**Table 5 ijms-26-02639-t005:** The adsorption thermodynamics of multiple heavy metals.

Heavy Metals	*C*_0_, mg·L^−1^	*T, K*	Δ*H*^0^, KJ·mol^−1^	Δ*S*^0^, JK^−1^ mol^−1^	Δ*G*^0^, KJ·mol^−1^
Pb(II)	50	298	1.08	36.52	−9.85
		308			−10.11
		318			−10.48
		328			−10.95
Cu(II)		298	19.01	80.76	−4.83
		308			−6.16
		318			−6.84
		328			−7.26
Cd(II)		298	18.02	66.91	−1.81
		308			−2.62
		318			−3.59
		328			−3.68
Zn(II)		298	22.95	78.66	−0.45
		308			−1.17
		318			−2.41
		328			−2.64

**Table 6 ijms-26-02639-t006:** A comparison of the adsorption capacities of modified UDLs with other adsorbents.

	Maximum Adsorption Capacity, *q_max_* (mg·g^−1^)				
Adsorbents	Pb(II)	Cu(II)	Cd(II)	Zn(II)	References
Chemically modified dragon fruit peel (DFP)	97.08	-	86.20	-	[39]
Incomplete incinerated *Urtica dioica* leaves (IIN)		46.47		-	[35]
Chemically modified orange peel	73.53	15.27	13.7	-	[61]
Kraft pulp-based carboxymethylated cellulose	20.3	7.8	-	8.4	[62]
Amino-modified wheat straw biochar	46.84	19.79	10.37	-	[51]
Commercial activated carbon (CGAC)	20.3	-	27.3	-	[16]
Unmodified *Urtica dioica* leaves (UDLs)	1.493	1.49	-	1.039	[36]
H_2_SO_4_-modified UDLs	98.3	73.4	12.9	13.9	In this study
TUDLs	46.6	37.1	11.9	10.6	In this study

## Data Availability

Data are contained within the article.

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
