# Peer review of "Modified Urtica dioica Leaves as a Low-Cost and Effective Adsorbent for the Simultaneous Removal of Pb(II), Cu(II), Cd(II), and Zn(II) from Aqueous Solution"

_ijms, 2025, doi:10.3390/ijms26062639_

Round 1
Reviewer 1 Report (Previous Reviewer 3)
Comments and Suggestions for Authors
The authors have done adequate corrections and the manuscript has been improved and can be published.
Author Response
"Please see the attachment."

Reviewer 2 Report (New Reviewer)
Comments and Suggestions for Authors
In this manuscript, the authors developed a cost-efficient adsorbent for metal ion removal based on the abundant Urtica dioica leaves. The H₂SO₄-modified leaves exhibit a high adsorption capacity of 98.3 mg/g, nearly five times that of commercial activated carbon. The authors have conducted a comprehensive study on the adsorption behavior and mechanism. The manuscript is well-written and suitable for publication in the International Journal of Molecular Sciences.
Below are some suggestions for the authors to consider:
-
The calcination of H₂SO₄-modified UDLs seems less relevant to this manuscript. Since calcination alters the molecular structure, the adsorption mechanism would differ from that of H₂SO₄-modified UDLs. Additionally, the authors mentioned that calcination could enhance adsorptive properties. It would be beneficial to include a direct comparison between UDLs and TUDLs in the main text to support this claim.
-
In Figure 9, only the black line in (a) and the red line in (b) have figure captions, which makes the labeling ambiguous. It would be clearer to remove them for consistency. Additionally, adding a data point between 0 and the first recorded point could provide stronger evidence that the adsorption of Pb and Cu follows the PSO relationship.
Author Response
"Please see the attachment."

This manuscript is a resubmission of an earlier submission. The following is a list of the peer review reports and author responses from that submission.
Round 1
Reviewer 1 Report
Comments and Suggestions for Authors
This article (manuscript no. 3308362) presents a new adsorbent for the simultaneous removal of Pb(II), Cu(II), Cd(II), and Zn(II) from aqueous solution. The adsorbent was derived from Urtica Dioica leaves via sulfuric acid treatment. The utilization of lignocellulosic materials, followed by their modification, enables the production of cost-effective and efficient adsorbents as alternatives to conventional, expensive adsorbents. The adsorbent was characterized using X-ray diffraction, Fourier transform infrared spectroscopy, scanning electron microscope, and X-ray photoelectron spectroscopy. This study determined the influence of adsorption time, solution pH, adsorbent dosage, temperature, and initial metal concentration on adsorption efficiency. Adsorption isotherms and kinetic curves were analysed using several models, and thermodynamic parameters of the adsorption were calculated. The topic is relevant to readers focused on pollutant removal, particularly heavy metal ions.
The manuscript is suitable for publication in the International Journal of Molecular Sciences, though I recommend minor revisions before publication.
Some of my comments are:
1) Line 141: Additional information about the “Pb²⁺, Cu²⁺, Cd²⁺, and Zn²⁺ complex solutions” should be included. It is necessary to specify why the chosen concentration range for metal ions is 50–500 mg/L, to substantiate the ratio of metal ions in the studied solutions. How do they relate to the actual composition of wastewater? What regularities might emerge if the metal ion ratios in the solution are altered?
2) Line 148: What is the loss of metal ions when filtering solutions through filter paper? What steps were taken to minimize this loss?
3) Table 4, 5: When analyzing adsorption isotherms and kinetic curves, it is insufficient to consider only the R² values. The error values between the experimental and model (calculated) adsorption values are also important.
4) Line 422: The statement, “According to the adsorption enthalpy and activation energy values, the adsorption processes are spontaneous and endothermic,” is unclear, as the manuscript does not include calculated adsorption activation energy values.
5) While the authors employed several valuable methods for studying the adsorbent, additional details on specific surface area and point of zero charge would be beneficial.
This article (manuscript no. 3308362) presents a new adsorbent for the simultaneous removal of Pb(II), Cu(II), Cd(II), and Zn(II) from aqueous solution. The adsorbent was derived from Urtica Dioica leaves via sulfuric acid treatment. The utilization of lignocellulosic materials, followed by their modification, enables the production of cost-effective and efficient adsorbents as alternatives to conventional, expensive adsorbents. The adsorbent was characterized using X-ray diffraction, Fourier transform infrared spectroscopy, scanning electron microscope, and X-ray photoelectron spectroscopy. This study determined the influence of adsorption time, solution pH, adsorbent dosage, temperature, and initial metal concentration on adsorption efficiency. Adsorption isotherms and kinetic curves were analysed using several models, and thermodynamic parameters of the adsorption were calculated. The topic is relevant to readers focused on pollutant removal, particularly heavy metal ions.
The manuscript is suitable for publication in the International Journal of Molecular Sciences, though I recommend minor revisions before publication.
Some of my comments are:
1) Line 141: Additional information about the “Pb²⁺, Cu²⁺, Cd²⁺, and Zn²⁺ complex solutions” should be included. It is necessary to specify why the chosen concentration range for metal ions is 50–500 mg/L, to substantiate the ratio of metal ions in the studied solutions. How do they relate to the actual composition of wastewater? What regularities might emerge if the metal ion ratios in the solution are altered?
2) Line 148: What is the loss of metal ions when filtering solutions through filter paper? What steps were taken to minimize this loss?
3) Table 4, 5: When analyzing adsorption isotherms and kinetic curves, it is insufficient to consider only the R² values. The error values between the experimental and model (calculated) adsorption values are also important.
4) Line 422: The statement, “According to the adsorption enthalpy and activation energy values, the adsorption processes are spontaneous and endothermic,” is unclear, as the manuscript does not include calculated adsorption activation energy values.
5) While the authors employed several valuable methods for studying the adsorbent, additional details on specific surface area and point of zero charge would be beneficial.
Reviewer 2 Report
Comments and Suggestions for Authors
Present paper reports the Adsorption performance of Urtica Dioica leaves for the simultaneous removal of Pb(II), Cu(II), Cd(II), and Zn(II) from aqueous solution. The manuscript requires major revision prior to publication. The comments to revise the manuscript are as follows:
1. There is not any caption of the first picture in line 119 or in the text, please correct.
2. Equation (1) must be corrected and rewrite “C” as “Ce”.
3. Equation (2) must be corrected and rewrite “Ci” as “Ce”.
4. Please, add EDS charts to the Fig.(2) or as supplementary file.
5. Please, add the % of Cd, Zn, Pb, and Cu to the Table 3 of XPS.
6. References must be added to each band in the FTIR results. The following paper can help you
DOI: 10.1080/03067319.2021.1943374; https://doi.org/10.1080/03067319.2019.1609462; https://doi.org/10.1007/s11356-022-23856-2; https://doi.org/10.1007/s10570-021-03749-2; DOI: 10.1080/01496395.2018.1445113.
7. In section 3.3.1; BET- analysis must be added to confirm increasing of surface area caused by sulfuric acid treatment.
8. In section 3.3.3; Zeta pot. Must be added to clarify the surface charge of the UDLs.
9. References must be added to all equation of kinetics and adsorption isotherms. This can help https://doi.org/10.1007/s10661-024-13142-8; https://doi.org/10.1007/s10967-020-07403-2; https://doi.org/10.1007/s10967-021-07730-y; https://doi.org/10.1007/s42452-019-1481-5.
10. Please check the qmax. (exp) of Cd and Zn in each kinetics and isotherms (Tables 4 and 5).
11. The papers of reference 25 and 29 must be added to the comparison with Table 7.
Reviewer 3 Report
Comments and Suggestions for Authors
The paper has presented the impact of acid modification on Urtica Dioica leaves for the removal of metal ions. Overall, though there are a lot of studies conducted, more discussions are missing. My specific comments are below.
1. Explain the statement and rationale "For the acid 222 concentration of 40%, more than 90% of all four heavy metals was adsorbed simultaneously, and when the acid concentration was greater than 40%, the adsorption amount de- 224 creased slowly." and also what are the possible reasons for Pb to better absorb and Zn to least adsorbed in the proposed adsorbent?
2. Why second order kinetics is showing the fit near 1? Any justifications? It is often suspected that due to the presence of time in both x and y axis, this could happen. Therefore, it is necessary to check with other forms of second-order kinetics. There are possibly 4. Please check the following studies for more clarification in the differences in second-order plots, when different equations are used. https://doi.org/10.1016/j.jcis.2017.09.055, https://doi.org/10.3390/su141711098
3. The authors have done isotherm, kinetics and thermodynamic analysis, but in my view due to accommodating all in one, the description became short. More explanations are required particularly with more relation and justification with the proposed adsorbent. For example, "Given the positive values of the change in enthalpy (∆H˚), the adsorption process was endothermic for multi-component heavy metals." But why is that, any explanations? "Adsorption 433 experiments revealed that the process followed Langmuir isotherms and pseudo-second- 434 order kinetics, indicating monolayer-adsorption and chemisorption mechanisms." why is that? And much more. Overall, much more critical discussions are needed.
4. The current Intro is more about the Urtica Dioica leaves that are used. But it should get an equal focus on the modification technique used and hypothesis behind this.
Round 2
Reviewer 2 Report
Comments and Suggestions for Authors
For my part, I have no further comments to make and the paper is suitable for publication.